# Synthesis and In Vitro Testing of YVO_4_:Eu^3+^@silica-NH-GDA-IgG Bio-Nano Complexes for Labelling MCF-7 Breast Cancer Cells

**DOI:** 10.3390/molecules28010280

**Published:** 2022-12-29

**Authors:** Tran Thu Huong, Le Thi Vinh, Hoang Thi Khuyen, Le Dac Tuyen, Nguyen Duc Van, Do Thi Thao, Ha Thi Phuong

**Affiliations:** 1Institute of Materials Science, Vietnam Academy of Science and Technology, 18 Hoang Quoc Viet, Hanoi 100000, Vietnam; 2Vietnam Academy of Science and Technology, Graduate University of Science and Technology, 18 Hoang Quoc Viet, Hanoi 100000, Vietnam; 3Faculty of Basic Science, Hanoi University of Mining and Geology, 18 Pho Vien, Hanoi 100000, Vietnam; 4Institute of Biotechnology, Vietnam Academy of Science and Technology, 18 Hoang Quoc Viet, Cau Giay, Hanoi 100000, Vietnam; 5Department of Chemistry, Hanoi Medical University, 1 Ton That Tung, Hanoi 100000, Vietnam

**Keywords:** YVO_4_:Eu^3+^, nanoparticles, bio-nanocomplexes, labelling, cancer cell labelling

## Abstract

We present a visual tool and facile method to detect MCF-7 breast cancer cells by using YVO_4_:Eu^3+^@silica-NH-GDA-IgG bio-nanocomplexes. To obtain these complexes, YVO_4_:Eu^3+^ nanoparticles with a uniform size of 10–25 nm have been prepared firstly by the hydrothermal process, followed by surface functionalization to be bio-compatible and conjugated with cancer cells. The YVO_4_:Eu^3+^@silica-NH-GDA-IgG nanoparticles exhibited an enhanced red emission at 618 nm under an excitation wavelength of 355 nm and were strongly coupled with MCF-7 breast cancer cells via biological conjugation. These bio-nanocomplexes showed a superior sensitiveness for MCF-7 cancer cell labelling with a detection percentage as high as 82%, while no HEK-293A healthy cells were probed under the same conditions of in vitro experiments. In addition, the detection percentage of MCF-7 breast cancer cells increased significantly via the functionalization and conjugation of YVO_4_:Eu^3+^ nanoparticles. The experimental results demonstrated that the YVO_4_:Eu^3+^@silica-NH-GDA-IgG bio-nanocomplexes can be used as a promising labelling agent for biomedical imaging and diagnostics.

## 1. Introduction

Nanostructured materials containing rare earth elements with numerous advantages such as high stability, strong luminescence, easy surface functionalization, and being friendly to environment and human body have been designed for new applications, especially for biomedical fluorescence labelling [1,2,3,4,5,6,7,8]. Most nanoparticles reported in past immunoassays are smaller than 200 nm in diameter for the biomolecular [9,10]. The size of these particles is proven to provide obvious prolonged equilibration time and enhanced nonspecific adsorption [11,12,13]. Therefore, the kinetic properties of smaller nanoparticles can be improved, and the nonspecific adsorption decreases. On the other hand, nanoparticles should be controlled to be big enough in size to bind with several proteins and cells on the surface, which is expected to increase the immunological affinity [14]. Among nanostructured materials containing rare-earth elements, YVO_4_:Eu^3+^ nanomaterials have received a great deal of interest because of biologically appropriate emission in the visible region and biocompatibility. It is surveyed from the literature that yttrium(III) orthovanadate, YVO_4_, is one of the most commonly used host lattices containing rare earth ions to prepare efficient luminescent materials with different color emittings because of its high luminescence quantum yields of *f–f* transitions [15]. Accordingly, when being doped with Eu^3+^ ions, YVO_4_:Eu^3+^ nanomaterials with high quantum efficiency have strong fluorescence emission at 618 nm. The luminescent 4*f*–4*f* transitions of europium(III) ion and the effective energy transfer from ligands to europium(III) ion lead to a strong emission of red light that makes them widely employed in detection, biomedical imaging, and luminescent labels [16,17].

The functionalization of nanostructured materials is a key step toward biomedical applications. The applications of nanostructured materials require preliminary grafting at the nanophosphor’s surface by organic or bio-organic functional groups. Different approaches are used such as encapsulation with functional polymers or direct grafting of biofunctional ligands. The 3-aminopropyl triethoxysilane (APTES) solution is well-known as a functionalized bio-compatible agent, because its ligands can create ethoxy groups on an inorganic surface. Meanwhile, glutaraldehyde (GDA) is one of the most popular bis-aldehyde homobifunctional crosslinkers that can be incorporated into nanostructured materials containing rare earth elements for many biomedical applications, including labelling, biosensing, drug delivery, and other therapies [18,19,20]. However, the structure of GDA is complicated, and its reaction mechanism is not fully understood. The reactions of GDA on proteins and other amine-containing molecules through the formation of a Schiff base that is lacking and still in progress [21]. In this work, we reported the results of the synthesis and in vitro testing of YVO_4_:Eu^3+^@silica-NH-GDA-IgG bio-nanocomplexes for labelling cancer cells.

## 2. Results and Discussion

### 2.1. Morphological Characterization

Figure 1a,b show the scanning electron microscopy (SEM) images of the YVO_4_:Eu^3+^ and YVO_4_:Eu^3+^@silica-NH-GDA-IgG nanoparticles, respectively. It indicated that YVO_4_:Eu^3+^ nanomaterials were nearly spherical with diameters ranging from 10 to 20 nm (Figure 1a). After being covered with silica and functionalized with APTES and IgG, the spherical nanoparticles with diameters increasing up to 20–25 nm were observed (Figure 1b).

### 2.2. Structure Characterization

Figure 2a shows the X-ray diffraction (XRD) pattern of the YVO_4_:Eu^3+^ sample. One can realize that all diffraction peaks of the sample coincided well with the standard data of the tetragonal-phase YVO_4_:Eu^3+^ structure (JCPDS No. 17-0341). Furthermore, no impurity peaks were detected, indicating that the dopant Eu^3+^ ions were well inserted into the host lattice of YVO_4_.

The characteristic chemical bonds of the YVO_4_:Eu^3+^ and YVO_4_:Eu^3+^@silica-NH-GDA-IgG samples were analyzed via Fourier-transform infrared (FTIR) spectra as shown in Figure 2b. For both cases, the observed absorption peaks at low frequencies of vibration such as at 644 and 790 cm^−1^ corresponded to the characteristic Y–O and Eu–O bonds, respectively. We also observed oscillations of the O–H bond at around 1638 and 3347 cm^−1^ [22]. In the case of YVO_4_:Eu^3+^@silica-NH-GDA-IgG nanoparticles, the band around 2880 cm^−1^ was attributed to C–H stretching vibration of alkanes. The N–H stretching vibration and the O–H bond were around 3347 cm^−1^. The characteristic band of the Si–O∓R bond was observed at around 1011 cm^−1^. Furthermore, an intense peak at 2360 cm^−1^ was associated with C=N stretching vibration that was formed by the reaction between the glutaraldehyde with amine–NH_2_ groups of functionalized YVO_4_:Eu^3+^@silica-NH-GDA-IgG antibodies. It indicated that the conjugation between luminescent nanoparticles and IgG was formed in the YVO_4_:Eu^3+^@silica-NH-GDA-IgG nanocomplex [20]. Compared to the case of the unconjugated YVO_4_:Eu^3+^ sample (Figure 2c) in addition to the strong peaks belonging to Y, V, and O elements, there were additionally weaker characteristic peaks of Eu and other peaks of Si, N, and C in the energy-dispersive X-ray (EDX) spectrum of the YVO_4_:Eu^3+^@silica-NH-GDA-IgG sample (Figure 2d). It confirmed that the silica shell and the amine group were successfully coated on the surface of the YVO_4_:Eu^3+^ core.

### 2.3. Luminescence Properties

Figure 3a shows the photoluminescence (PL) spectra of the YVO_4_:Eu^3+^, YVO_4_:Eu^3+^@silica-NH_2_, and YVO_4_:Eu^3+^@silica-NH-GDA-IgG samples at a 355 nm excitation wavelength. The PL spectra consisted of a narrow band corresponding to the well-known Eu^3+^ emission from intra 4*f* transitions (^5^D_0_–^7^F_1_, ^5^D_0_–^7^F_2_, ^5^D_0_–^7^F_3_, and ^5^D_0_–^7^F_4_). The strongest emission peaks were yielded by the ^5^D_0_–^7^F_2_ transition at around 618 nm (red light). Specially, the positions of PL peaks of the YVO_4_:Eu^3+^@silica-NH-GDA-IgG sample were almost the same as those of the YVO_4_:Eu^3+^ sample and the YVO_4_:Eu^3+^@silica-NH_2_ sample. It implies that the PL property of the YVO_4_:Eu^3+^@silica-NH-GDA-IgG sample was unchanged after functionalization and biological conjugation. Moreover, the PL intensity of the YVO_4_:Eu^3+^@silica-NH-GDA-IgG sample was enhanced. This result can be explained by substitution of the O–H luminescence quenching with O=C and C–H in the case of surface modification. These fluorescence properties of samples containing YVO_4_:Eu^3+^ nanoparticles have attracted a great deal of attention in biology and medicine. It can be indicated that the surface modification of the YVO_4_:Eu^3+^ nanoparticles not only improved their bio-compatibility, but also increased their PL intensity. Furthermore, the YVO_4_:Eu^3+^@silica-NH-GDA-IgG sample remained stable for quite a long time. The PL intensity of the YVO_4_:Eu^3+^@silica-NH-GDA-IgG sample had insignificant changes after three and six months of the synthesis as shown in Figure 3b. Thus, these nanoparticles can be used for bio-labelling applications.

### 2.4. In Vitro Cellular Imaging

We used fluorescence microscopy to evaluate the linking ability between YVO_4_:Eu^3+^@silica-NH-GDA-IgG conjugates and MCF-7 breast cancer cells after the incubation process. Figure 4a–c show the fluorescent images for three cases of MCF-7 breast cancer cells (negative control), incubated MCF-7 breast cancer cells with YVO_4_:Eu^3+^@silica-NH_2_ nanoparticles, and incubated MCF-7 breast cancer cells with YVO_4_:Eu^3+^@silica-NH-GDA-IgG nanoparticles, respectively.

For the first case, we did not observe PL emission in the reference sample (Figure 4a). In the second case, the incubated MCF-7 breast cancer cells with the YVO_4_:Eu^3+^@silica-NH_2_ sample showed a blur and tiny PL intensity (Figure 4b). This can be explained by the existence of the weak bonds between YVO_4_:Eu^3+^@silica-NH_2_ nanoparticles and the cancer cells. As reported, muscarinic acetylcholine receptors (mAChR) belong to the G-protein-coupled receptor family and are extensively expressed in human breast tumor cells. In addition, immunoglobulin G (IgG) has been described that the presence of IgG in tumor cells establishes correlations between high antibody levels and promotion of cancer cell proliferation, invasion, and poor clinical prognosis for tumor patients. Blocking tumor-cell-derived IgG inhibits tumor cells. Tumor-cell-derived IgG might impede antigen-dependent cellular cytotoxicity by binding antigens such as mAChR while, at the same time, lacking the capacity for complement activation [23]. However, we can observe bright red pixels in the case of YVO_4_:Eu^3+^@silica-NH-GDA-IgG nanoparticles in Figure 4c. This evidence demonstrated a strong coupling between YVO_4_:Eu^3+^@silica-NH-GDA-IgG nanoparticles and MCF-7 breast cancer cells due to biological conjugation.

Furthermore, it can be seen that YVO_4_:Eu^3+^@silica-NH-GDA-IgG nanoparticles were localized within the cell cytoplasm. The high GDA concentration allowed the ligand binding between cells and luminescent labelling particles. After that, the YVO_4_:Eu^3+^@silica-NH-GDA-IgG nanoparticles were internalized into the cell via the invagination process. Therefore, YVO_4_:Eu^3+^@silica-NH-GDA-IgG nanoparticles could be used as a potential bio-label for MCF-7 breast cancer cells. In comparison to the other techniques, the use of YVO_4_:Eu^3+^@silica-NH-GDA-IgG nanoparticles is expected to be more advantageous by providing a facile method without requirements of any complex apparatus as well as processing such as spectral equipment and data analysis [24,25]. Additionally, it is a visual tool that is needed for some specific studies in biology.

In addition, the nanocomplex exhibited much fewer probing activities on HEK-293A human embryonic kidney cells, which were non-cancerous cell lines. Figure 5a–c show the fluorescence images of the HEK-293A cells with YVO_4_:Eu^3+^@silica-NH-GDA-IgG nanoparticles with different detection modes: (a)—negative control, (b)—Dark field, (c)—Merge, respectively. The experimental conditions were the same as those for the MCF-7 breast cancer cells, and the images were taken in the cases of the bright field, dark field, and merged modes. Moreover, the YVO_4_:Eu^3+^@silica-NH-GDA-IgG nanoparticles could not probe healthy cells of HEK-293A.

The flowcytometry results also provided the percentage of probed cells using the YVO_4_:Eu^3+^@silica-NH-GDA-IgG nanocomplex. As shown in Figure 6, YVO_4_:Eu^3+^@silica-NH-GDA-IgG nanocomplexes were found in about 82.11% of MCF-7 cells when stained with YVO_4_:Eu^3+^@silica-NH-GDA-IgG nanocomplexes. A similar result (1.55%) was found in MCF-7 cells that were incubated with YVO_4_:Eu^3+^@silica-NH_2_ nanoparticles. The percentage of MCF-7 cells was only 0.57% for MCF-7 cells incubated with unconjugated YVO_4_:Eu^3+^ nanoparticles. Thus, the detection percentage of MCF-7 breast cancer cells increased pronouncedly from 0.57% for the case of unconjugated YVO_4_:Eu^3+^ nanoparticles to 1.55% for the case of YVO_4_:Eu^3+^@silica-NH_2_ nanoparticles and achieved a highest value of 82.11% with YVO_4_:Eu^3+^@silica-NH-GDA-IgG nanoparticles, indicating the crucial role of the functionalization and conjugation of YVO_4_:Eu^3+^ nanoparticles.

## 3. Materials and Methods

### 3.1. Preparation of YVO_4_:Eu^3+^@Silica-NH_2_ Nanoparticles

In a typical synthesis of YVO_4_:Eu^3+^ nanoparticles, yttrium nitrate hexahydrate (Y(NO_3_)_3_∙6H_2_O, 99.9%; Sigma-Aldrich), sodium orthovanadate (Na_3_VO_4_ 90%; Sigma-Aldrich), and europium nitrate pentahydrate (Eu(NO_3_)_3_∙5H_2_O, 99.9%; Sigma-Aldrich) were mixed with a molar ratio of 0.99/1/0.01, and sodium hydroxide (NaOH 99%, Merck) was added to control pH values of 6–8. The solution was stirred for 180 min, and then, it was transferred into an autoclave and heated at 190 °C for 24 h. The YVO_4_:Eu^3+^ nanoparticles were separated by using centrifugation (5800 rpm) and washed with deionized water for three times and dispersed in ethanol to achieve a colloidal solution. YVO_4_:Eu^3+^ nanoparticles were coated by silica via the Stöber process. In detail, 10 mL of tetraethyl orthosilicate (TEOS), 10 mL of ethanol (NH_4_OH), 1 mL of acetic acid (CH_3_COOH), and 2 mL of deionized water (H_2_O) were mixed and stirred at room temperature for 24 h. Ten millimeters of the colloidal solution of YVO_4_:Eu^3+^ nanoparticles was slowly added into the above mixture and continuously stirred for 24 h. The YVO_4_:Eu^3+^@silica nanoparticles were separated by using a centrifuge and dispersed in 20 mL of ethanol. Then, they were added into a mixture of 22.5 mL absolute ethanol and 2 mL 3-aminopropyltriethoxysilane (APTES) at 60 °C for 5 h. YVO_4_:Eu^3+^@silica-NH_2_ nanoparticles were obtained by centrifugation [26,27,28].

By using APTES, a reagent contains a short organic 3-amino propyl group, which terminates in a primary amine, and ethoxy groups are not reactive enough to couple spontaneously with OH groups on an inorganic surface without prior hydrolysis to make silanol. This protocol can be used to modify the surface of particles with this reagent as described in Figure 7. Firstly, the reaction involves the hydrolysis of the alkoxysilane group to create highly reactive silanol that undergoes hydrogen bonding with other silanol groups in solutions and on the particle surface, resulting in the associated organosilane derivatives. Then, a condensation reaction takes place to form a polymerized coating of the organosilane on the particle surface [21]. Secondly, APTES is coated on the YVO_4_:Eu^3+^@silica surface to create a covalent shell containing a primary amine group. The reaction occurs in a partially aqueous environment, because ethoxy groups are unreactive enough to substrate OH groups without prior hydrolysis. This is typically performed in 5% water in ethanol that is acidified with acetic acid to pH values of 4.5–5.5. The process results in a layer containing about 3–8 organosilanes in thickness and masks the inorganic substrate with aminopropyl groups. The advantage of this process is providing a thin and controllable silane layer that can be created a monolayer of the aminopropyl group on the surface.

### 3.2. Preparation of YVO_4_:Eu^3+^@silica-NH-GDA-IgG Bio-Nanocomplexes

Figure 8 describes the functionalization of the YVO_4_:Eu^3+^@silica-NH nanoparticles with glutaraldehyde (GDA) and immunoglobulin G (IgG). The YVO_4_:Eu^3+^@silica-NH_2_ nanoparticles and GDA were dispersed in vanadate-buffered saline (PBS, 0.1 M, pH 5) with a concentration of 5 gL^−1^. Then, this compound was added to different concentrations of IgG. These reaction mixtures were incubated with glycerol at room temperature for 4 h. The YVO_4_:Eu^3+^@silica-NH-GDA-IgG products were collected by centrifugation (5900 rpm) with water for three times and stored at 4 °C in a closing box. The reaction mechanism can be explained via a Schiff base linkage with amine on proteins [21]. Proteins can be coupled to the -NH_2_ groups via an amine-reactive linker—glutaraldehyde—to form activated derivatives that enable to make crosslink with other proteins. As shown in Figure 8, the amine groups react with the aldehyde groups to form a Schiff base, resulting in a polymeric coating that contains both aldehydes and double bonds for further coupling with amine-containing molecules.

### 3.3. MCF-7 Breast Cancer Cell and HEK-293A Cell Culture and Fluorescence Imaging of Cells

In this study, the experiments were implemented on MCF-7 breast cancer cells (MCF-7 was kindly presented by Prof. Chi-Ying Huang, National Yangming University, Taiwan-MCF7 (ATCC # HTB-22). The results were then compared with HEK-293A healthy cells (these cells were kindly provided from Prof. Young-Pil Kim, HanYang University, Korea-HEK-293A (Invitrogen # P/N 51-0036)) that were maintained in a cultured medium—Dulbecco’s Modified Eagle Medium (DMEM) with fetal bovine serum (10%) (Sigma) and gentamicin (50 µg/mL) at 37 °C and 5% CO_2_ in a humidified atmosphere [29,30]. The cells were seeded at a density of 5.10^4^ cells/mL. To study the uptake capacity of the YVO_4_:Eu^3+^@silica-NH-GDA-IgG nanoparticles, the MCF-7 breast cancer cells and HEK-293A cells (10^6^ cells/mL) at the log phase were seeded in 24-well plates and then incubated for 24 h. The YVO_4_:Eu^3+^@silica-NH-GDA-IgG nanoparticles (with a concentration of 20 µg/mL) were then added to the cell-seeded wells for 3 h. After the assigned time, the cultured medium was discarded. The cells were then washed three times with phosphate-buffered saline. At the end of the process, phosphate-buffered saline was added to the wells, before and after MCF-7 breast cancer cells and HEK-293A cells were incubated with YVO_4_:Eu^3+^@silica-NH-GDA-IgG bio-nanocomplexes. The cell images were obtained using an Olympus ScanR 100X fluorescent microscope.

### 3.4. Cellular Surface Labelling Analysis Using Flowcytometry

MCF-7 cancer cells, a cultured medium (DMEM), a fetal bovine serum (FBS), trypsin-EDTA, and bovine insulin were obtained from Invitrogen (Carlsbad, CA, USA). MCF-7 cells (5 × 10^4^ cells/mL) were cultured in 6-well plates at 37 °C and 5% CO_2_ for 24 h. Then, the cells were detached with 0.05% trypsin-EDTA and centrifuged at 1000 rpm for 5 min to obtain the cell pellets. The cells were fixed with 4% formaldehyde for 24 h at 4 °C and washed with cold phosphate-buffered saline (PBS). To access cellular surface labeling, the YVO_4_:Eu^3+^-NH_2_ and YVO_4_:Eu^3+^@silica-NH-GDA-IgG nanoparticles were employed and incubated with fixed cells for 2 h. The labelled cells were washed with cold PBS twice before resuspending in PBS for analyzing with a flowcytometry Novocyte system (ACEA Bioscience inc.) and NovoExpress software. The cells were requested for light protection.

### 3.5. Characterization Techniques

The X-ray diffraction (XRD) analysis of the samples was carried out on a Siemens D5000 diffractometer with λ = 1.5406 Å. The morphologies and the energy-dispersive X-ray spectra were observed and measured by field emission scanning electron microscopy (S-4800; Hitachi) attached with an energy-dispersive X-ray spectrometer. The infrared absorption spectra were performed by employing a Fourier-transform infrared spectrometer (FTIR NEXUS 670). The photoluminescence (PL) spectra of the samples were studied by using an iHR550 photoluminescence system (Horiba). The cells were observed under an Olympus ScanR fluorescence microscope (Olympus Europa SE & Co.KG, Hamburg, DE).

## 4. Conclusions

YVO_4_:Eu^3+^ nanoparticles with a uniform size of 10–25 nm were synthesized via a hydrothermal process, followed by further functionalizations to form the YVO_4_:Eu^3+^@silica-NH-GDA-IgG bio-nanocomplexes. The surface modification of the YVO_4_:Eu^3+^ nanoparticles not only improved the bio-compatible media, but also increased their PL intensity due to enhancement of chemical stability. The experimental evidence indicated that the YVO_4_:Eu^3+^@silica-NH-GDA-IgG nanoparticles could selectively detect MCF-7 breast cancer cells while it could not probe HEK-293A healthy cells for in vitro tests. The YVO_4_:Eu^3+^@silica-NH-GDA-IgG bio-nanocomplexes with a significant bio-compatible capability exhibited a strong, enhanced red emission, which provides a visual tool for bio-labelling applications.

## Figures and Tables

**Figure 1 molecules-28-00280-f001:**
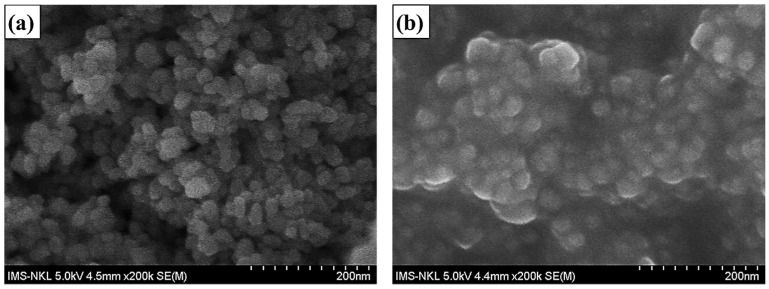
SEM images of the YVO_4_:Eu^3+^ sample (**a**) and YVO_4_:Eu^3+^@silica-NH-GDA-IgG (**b**) samples.

**Figure 2 molecules-28-00280-f002:**
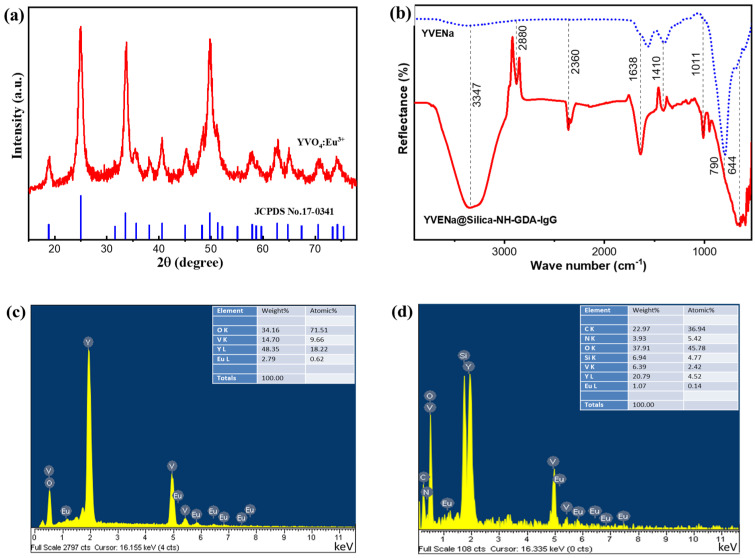
(**a**) X-ray diffraction pattern of the YVO_4_:Eu^3+^ sample in comparison with the standard data of tetragonal-phase YVO_4_:Eu^3+^ nanoparticles (JCPDS No. 17-0341) as a reference. (**b**) FTIR spectra of the YVO_4_:Eu^3+^@silica-NH-GDA-IgG sample and the YVO_4_:Eu^3+^ samples. (**c**) EDX spectrum of the YVO_4_:Eu^3+^ sample. (**d**) EDX spectrum of the YVO_4_:Eu^3+^@silica-NH-GDA-IgG sample.

**Figure 3 molecules-28-00280-f003:**
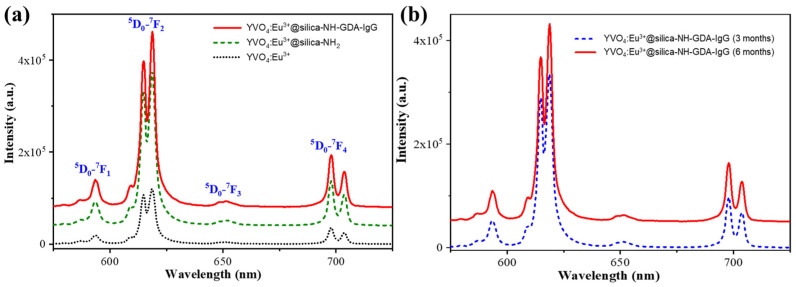
(**a**) PL spectra of the YVO_4_:Eu^3+^, YVO_4_:Eu^3+^@silica-NH_2_, and YVO_4_:Eu^3+^@silica-NH-GDA-IgG nanoparticles under a 355 nm excitation. (**b**) PL spectra of the YVO_4_:Eu^3+^@silica-NH-GDA-IgG nanoparticles measured after three and six months of the synthesis.

**Figure 4 molecules-28-00280-f004:**
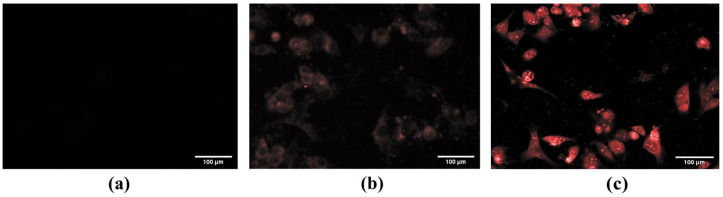
Fluorescent microscopy images of MCF-7 breast cancer cells after 3 h of incubation without nanoparticles (negative control) (**a**) and with YVO_4_:Eu^3+^@silica-NH_2_ nanoparticles at a concentration of 20 µg/mL (**b**) and YVO_4_:Eu^3+^@silica-NH-GDA-IgG conjugates at a concentration of 20 µg/mL (**c**) under an excitation wavelength in the UV region.

**Figure 5 molecules-28-00280-f005:**
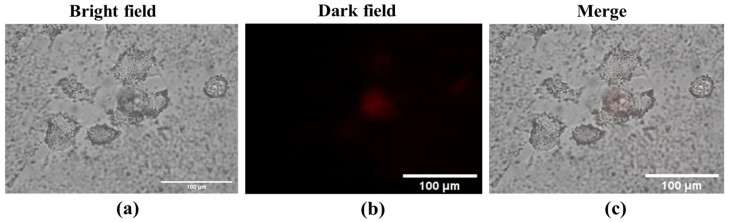
Fluorescence images of HEK-293A embryonic kidney cells after incubated with YVO_4_:Eu^3+^@silica-NH-GDA-IgG bio-nanocomplexes with different detection modes: (**a**)—negative control, (**b**)—Dark field, (**c**)—Merge.

**Figure 6 molecules-28-00280-f006:**
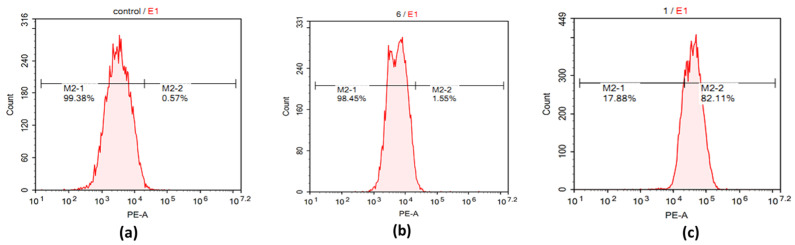
Flowcytometry analysis of labelled MCF-7 cells incubated with the negative control (**a**), YVO_4_:Eu^3+^@silica-NH_2_ nanoparticles (**b**), and YVO_4_:Eu^3+^@silica-NH-GDA-IgG nanoparticles (**c**).

**Figure 7 molecules-28-00280-f007:**
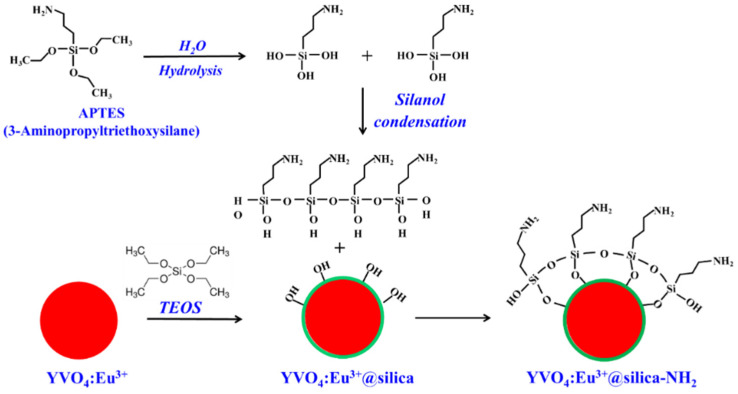
The coupling reaction of the silanol group to the YVO_4_:Eu^3+^ nanoparticles and APTES coating on the YVO_4_:Eu^3+^@silica surface to create a covalent shell of the amine group.

**Figure 8 molecules-28-00280-f008:**
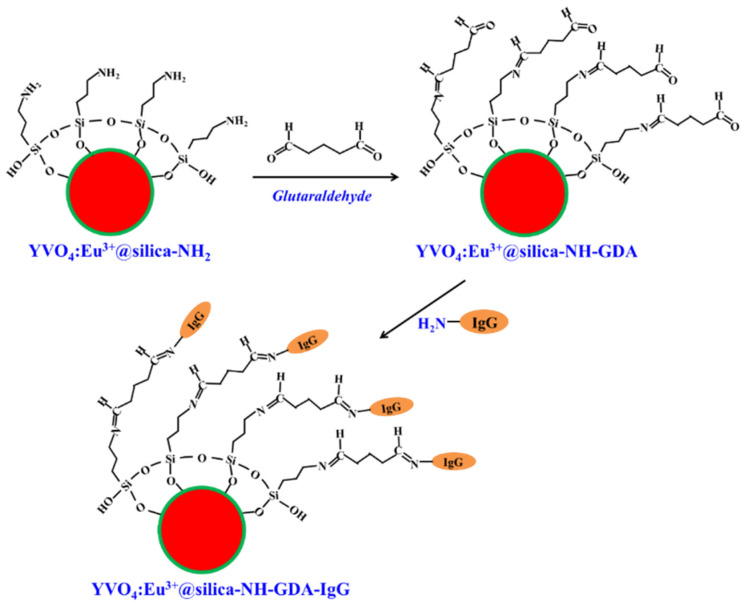
Functionalization of the YVO_4_:Eu^3+^@silica-NH nanoparticles with GDA and IgG. This conjugation strategy has been used to associate biomolecules containing amine groups with aminated YVO_4_:Eu^3+^@silica-NH_2_ nanomaterials, usually utilizing glutaraldehyde, a molecule containing two aldehyde moieties. One aldehyde group forms a Schiff base with the amine groups, while the other binds to the amino groups of the biomolecules. The reactions with glutaraldehyde are favored in alkaline media, being more efficient at high pH values [21].

## Data Availability

All data are available in this publication.

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
