# Peer review of "Synthesis and In Vitro Testing of YVO4:Eu3+@silica-NH-GDA-IgG Bio-Nano Complexes for Labelling MCF-7 Breast Cancer Cells"

_molecules, 2022, doi:10.3390/molecules28010280_

Round 1

Reviewer 1 Report

The authors presented a research article on labeling MCF-7 breast cancer cells with bioconjugated rare-earth nanoparticles. The YVO4:Eu3+ nanoparticles were functionalized with APTES and conjugated with IgG for specific labeling of MCF-7 cells with HEK-293A cells as a control. Although the result suggested high detection percentage and specificity with the nanocomplex developed in this study, comprehensive characterization of the material is lacking. The author should resubmit the article with the following comments addressed.

  1. IgG is a class of antibodies. What is the specific antigen that the IgG moiety used in this study targets on MCF-7 cells but not HEK-293A cells? The author should describe the rationale of selection of targeting moiety.

  2. The article claimed that the size of the YVO4:Eu3+ nanoparticles are 10-20 nm and with bioconjugation the size increased to 20-25 nm based on SEM images. However, it is hard to tell if the nanoparticles are monodispersed stable colloids based on the SEM images. Extended characterization on the size of the nanoparticles, such as DLS or NTA, should be provided.

  3. FTIR spectra in Figure 2b showed a peak at 2360 cm-1 associated with C=N stretching vibration and the author claimed that the peak indicated conjugation of IgG on nanoparticles. However, the peak could only indicate a successful reaction of glutaraldehyde to YVO4:Eu3+@silica-NH2. Additional evidence of IgG conjugation should be provided, such as immunostaining, or surface charge or hydrophobicity change pre- and post-conjugation.

  4. Fluorescence microscopy images of MCF-7 and HEK cells in Figure 4 and Figure 5 indicated specificity of labeling MCF-7 breast cancer cells. Scale bars should be added, and providing images with the same magnification for MCF-7 and HEK cells is suggested. Besides, flow cytometry data for MCF-7 and HEK cells after incubation with the nanocomplex should be provided as additional evidence for specificity.

  5. On Page 8, Line 234, PEG1500 was added together with the nanocomplex for cell labeling. What is the role of PEG1500 for the labeling? On Page 8, Line 237, why is centrifugation used to discard the medium from the plate? The medium could simply be removed by aspiration as the cells are adhesive. 

  6. On Page 9, Line 251, for flow cytometry, the nanocomplex was incubated with fixed cells for labeling. Is there a reason why fixed cells are used for labeling rather than live cells? The author could label live cells with the nanocomplex and then harvest the cells by trypsinization for flow cytometry. What is the detecting percentage and cell-associated fluorescence for live cells labeling with control, unconjugated and IgG-conjugated nanoparticles? 

Author Response

Dear Madam/Sir,
I and all my co-workers would like to thank you in deep for careful reading and very useful comments that help us to complete our manuscript with the article entitled: “Synthesis and in vitro testing of YVO4:Eu3+@silica-NH-GDA-IgG bio-nano complexes for labelling MCF-7 breast cancer cells”. We have already revised and returned our manuscript to the Molecules 's office. However, we would like to discuss in details all your comments as follows "Please see the attachment"

Sincerely,

Tran Thu Huong, Ph.D.

Reviewer 2 Report

This word developed a visual tool for labelling MCF-7 breast cancer cells by using YVO4:Eu3+@silica-NH-GDA-IgG nanoparticles. The YVO4:Eu3+@silica-NH-GDA-IgG nanoparticles shows strong selectively on cancer cells compared to particles without surface modification. However, there are some problems need to be addressed and several improvements can be made. Here are my comments:

1: Language need to be improved. Grammar errors (such as page 2, line 57, etc.) need to be corrected.

2: Images in this draft need to be prepared in a better way.

(1)    For Figure 1, the images are not well focused. Why the voltage used for Figure 1 a and Figure 1 b are different?

(2)    The Eu signal shown in Figure 2 c are too weak, which may not be adequate for proving the present of Eu element. Longer acquisition time for EDS test or testing on a zoomed in region are suggested.

(3)    The explanation on Figure 5 can be improved. Hek-293A embryonic kidney cells can be identified with arrows in the figure a. Is the central red spot shown in Figure 5 b the fluorescence of nanoparticles or the incident light?

(4)    The font size in different figures is not uniform.

3: The importance of particle size was discussed in the introduction. However, in the results and discussion are not well discussed.

(1)    After the surface decoration, the particles seem to aggregate with each other and covered by polymer, as shown in Figure 2. How do you determine the diameter of decorated particles?

(2)    Is there a particular reason to choose the 10-20 nm particles?

4: It was claimed that nanoparticles were localized within the cell. Is there any evidence prove that the nanoparticles located inside the cell instead of on the cell membrane?

5: To verify the selectivity of YVO4:Eu3+@silica-NH-GDA-IgG, HEK-293A embryonic kidney cells was used to compare with MCF-7 breast cancer cells. Is there a particular reason to use kidney cells?

6: Current PL tests may not be able to provide enough evidence for chemical stability of surface modified nanoparticles. To verify chemical stability, further experiments about PL intensity in relation with time are needed.

I suggest a major revision for this paper.

Author Response

(The authors gave the same response as above.)

Round 2

Reviewer 1 Report

The authors have addressed the majority of the comments. However, for Comment 1, the authors still did not describe which antigen on MCF-7 cells that the selected IgG antibody binds to. More information on the selectivity of the IgG antibody and the corresponding antigen should be provided. 

Author Response

Dear Madam/Sir,

I and all my co-workers would like to thank you in deep for careful reading and very useful comments that help us to complete our manuscript with the article entitled: “Synthesis and in vitro testing of YVO4:Eu3+@silica-NH-GDA-IgG bio-nano complexes for labelling MCF-7 breast cancer cells”. We have already revised and returned our manuscript to the Molecules 's office. However, we would like to discuss in details all your comments as follows "Please see the attachment":

Reviewer 2 Report

The authors did a good job in addressing most of the comments. 

However, I have two more concerns about this manuscript.

1: I suggest adding the results of “Photoluminescence spectra of the YVO4: Eu3+ @silica-NH-GDA-IgG nanoparticles with 3 months and 6 months” or more detailed results in figure 3.  It is not proper to directly claim “coating layers might enhance the chemical stability of the resulting complexes”(line 117) without experimental result support.

2: In figure 5, there are six cells could be identified. However, from figure 5 b and c, it seems that nanoparticles only incubated in the central cell or only the central cell shows florescence. If this is the case, please explain it. If all six cells show fluorescence, please adjust the brightness/contrast. 

Author Response

(The authors gave the same response as above.)
